# Label-Free DNA Detection Using Etched Tilted Bragg Fiber Grating-Based Biosensor

**DOI:** 10.3390/s23167019

**Published:** 2023-08-08

**Authors:** Abdullah Al Noman, Jitendra Narayan Dash, Md Abdullah Al Maruf, Cheng Xin, Hwa-Yam Tam, Changyuan Yu

**Affiliations:** 1Department of Electronic and Information Engineering, Photonics Research Institute, The Hong Kong Polytechnic University, 11 Yuk Choi Rd, Hung Hom, Hong Kong SAR, China; abdullahal.noman@polyu.edu.hk; 2Department of Electrical Engineering, Photonics Research Institute, The Hong Kong Polytechnic University, 11 Yuk Choi Rd, Hung Hom, Hong Kong SAR, China; jnphysics@gmail.com (J.N.D.); hwa-yaw.tam@polyu.edu.hk (H.-Y.T.); 3Department of Health Technology and Informatics, The Hong Kong Polytechnic University, 11 Yuk Choi Rd, Hung Hom, Hong Kong SAR, China; abdullah.maruf@polyu.edu.hk

**Keywords:** optical fiber sensor, tilted fiber Bragg grating, etching, DNA, biosensor

## Abstract

A label-free-based fiber optic biosensor based on etched tilted Bragg fiber grating (TFBG) is proposed and practically demonstrated. Conventional phase mask technic has been utilized to inscribe tilted fiber Bragg grating with a tilt angle of 10°, while the etching has been accomplished with hydrofluoric acid. A composite of polyethylenimine (PEI)/poly(acrylic acid) (PAA) has been thermally deposited on the etched TFBG, followed by immobilization of probe DNA (pDNA) on this deposited layer. The hybridization of pDNA with the complementary DNA (cDNA) has been monitored using wavelength-dependent interrogation. The reproducibility of the probes has been demonstrated by fabricating three identical probes and their response has been investigated for cDNA concentration ranging from 0 μM to 3 μM. The maximum sensitivity has been found to be 320 pm/μM, with the detection limit being 0.65 μM. Furthermore, the response of the probes towards non-cDNA has also been investigated in order to establish its specificity.

## 1. Introduction

Biosensors have garnered immense attention over the course of the last few years due to their extensive usage in ecology, food quality assessment, human body and medical disorder analysis [1,2,3,4,5]. In particular, deoxyribonucleic acid (DNA)-based biosensors are deemed to be the most frequently utilized sensors in various scientific domains (e.g., medical, drug, epidemic prevention and genetic) [6,7,8,9,10]. Furthermore, conventional approaches such as electrochemical [11] and fluorescent [12] methods have been demonstrated to monitor DNA. However, these methods are constrained by various issues fluorescence-based devices often involve a UV light source to excite the fluorescence and label-based complementary DNA (cDNA) to attach to the probe, leading to high cost and restricted fluorescence lifespan [13,14]. Thus, these issues can be addressed by the label-free-based optical fiber sensors (OFSs) as they offers numerous benefits, such as immune to electromagnetic interference, compactness, minimal cost and high precision over traditional techniques [15,16,17].

Various fiber optic sensors relying on different methods have been reported to identify DNA thus far, including surface plasmon resonance (SPR) [18], Fabry–Pérot interferometer (FPI) [19], Mach–Zehnder interferometer (MZI) [20], etc. SPR-dependent biosensors require metal coating with precise thickness and interferometric biosensors exhibit low mechanical strength as well as a spectrum with low amplitude [21,22]. Meanwhile, many fiber grating-based devices (FBDs) have been regarded as suitable candidates for DNA detection due to their high sensitivity, high reproducibility and multi-parameter sensing capabilities [23]. For instance, a poly-L-lysine (PLL)-coated long fiber grating (LPG) has been proposed to trace DNA where the red shift was seen because of the attachment between PLL/pDNA (probe DNA) and cDNA [24]. The sensor displayed an average sensitivity of 1.82 nm upon DNA hybridization. Another LPG-based sensor using modified pDNA with 1-ethyl-3-(3-dimethylaminopropyl) carbodiimide (EDC) has been reported for a similar application [25]. However, the resonant dip was found to be unstable after one hour of the detection period and the detection limit was calculated as 4 nM. In addition, microfibers have been applied for sensing applications due to their strong evanescent field characteristics. A microfiber fiber Bragg grating (mFBG)-coated PLL layer has been demonstrated for DNA sensing with a sensitivity of 0.456 nm/μM and a minimum detection limit of 0.5 μM DNA [26]. A similar approach has been followed using layer-by-layer self-assembly technology to detect cDNA [27]. Additionally, the sensor has been found to have a sensitivity response of around 200 pm for a 1 μM change in concentration of cDNA. Nonetheless, all the aforementioned sensors involved complex fabrication processes and they require an additional fiber Bragg grating (FBG) for temperature or strain monitoring to enhance the device’s accuracy. These concerns can be resolved through the use of tilted fiber Bragg grating (TFBG), where the cladding modes can be utilized to monitor change in the ambient environment while the Bragg wavelength can be used to monitor other parameters. For example, a number of RI-based sensors employing TFBG in intensity [28], TFBG interaction with multimode fiber [29] and Brillouin scattering in TFBG [30] have been demonstrated. Moreover, a TFBG sensor built on SPR technic has been utilized for DNA hybridization by applying a graphene/silver membrane and the lowest detection threshold of the sensor has been found to be 3 pM [31].

In this paper, we propose an etched TFBG-based sensor to detect DNA hybridization. The device is coated with a mixture of polyethylenimine (PEI) and poly(acrylic acid) (PAA) layer, followed by a specific pDNA attachment for detecting different cDNA concentrations. PEI is an economical chemical that easily forms bonding with DNA through negative charge interactions [32]. The binding between the pDNA and cDNA has been observed using wavelength-based interrogation. Furthermore, the specificity of the sensor has been verified by monitoring its interaction with non-complementary DNA (non-cDNA) for a certain period. Additionally, the minima corresponding to Bragg resonance has also been monitored to minimize the error due to other parameters.

## 2. Sensing Principle, Fabrication and Performance of TFBG Length

### 2.1. Sensing Principle

The schematic of a typical TFBG structure is displayed in Figure 1. Similar to conventional FBG, the Bragg resonance in the TFBG core forms as the light coupling occurs in the forward- and counter-propagating core modes. However, the presence of tilted planes results in the excitation of cladding modes and the interaction between the forward-propagating core modes and the counter-propagating cladding modes leads to resonance. The core-cladding mode resonance is usually analyzed in transmission mode owing to the significant attenuation of cladding modes in reflection mode. Furthermore, the evanescent field of the guided modes in cladding interacts with the ambient environment, thereby changing their effective indices and subsequently changing the resonance spectrum. The resonance wavelength of the Bragg condition ${(\lambda }_{Bragg})$ and cladding modes ($\lambda_{cl}^{m}$) can be as expressed as in [33].
(1)$$\lambda_{Bragg}=\frac{2n_{eff,cr}\Lambda }{\cos\theta }$$
(2)$$\lambda_{cl}^{m}=\frac{(n_{eff,cr}+n_{eff,cl}^{m})\Lambda }{\cos\theta }$$
where $n_{eff,cr}$ and $n_{eff,cl}^{m}$ refer to the effective refractive index of the core mode at $\lambda_{Bragg}$ and *m*th cladding mode at $\lambda_{cl}^{m}$, respectively. Due to the tilt in the grating planes along the fiber axis, the grating period can be written as $\Lambda_{g}=\Lambda /\cos\theta$, where *θ* represents the tilted angle of the grating planes with the fiber axis.

### 2.2. TFBG Fabrication, Experimental Setup and Response towards Change in RI

The TFBG was photoinscribed in a single-mode fiber (SMF) using a UV laser (On-Tech, λ = 213 nm, energy = 6 μJ)-based phase mask procedure. In this process, the photosensitivity of the fiber was improved by storing it in a hydrogen-sealed container for a week at a temperature of 75 °C and a pressure of 1300 psi prior to the inscription. A line-focused laser beam passing through a diffractive phase mask (pitch = 1071.77 nm) mounted on a rotational stage was incident onto the hydrogen-adapted fiber. During the inscribing process, the phase mask was slanted at an angle of 10 degrees with respect to the fiber axis and the resultant TFBG spectrum was monitored using an optical spectrum analyzer (OSA, AQ6370D, YOKOGAWA, Tokyo, Japan). Thereafter, the TFBG-inscribed fiber was kept in a temperature chamber at 70 °C for 12 h to remove any residual hydrogen, thereby improving its stability. Subsequently, the fiber was etched (see Figure 2a) by immersing it in a solution of 48% hydrofluoric acid (HF) for 30 min to enhance the sensitivity towards the ambient environment. Afterwards, the fiber was bathed with DI water to remove the remaining HF from the fiber. The diameter of the etched fiber was found to be about 40 μm, as shown in Figure 2a.

The experimental setup consists of a homemade broadband source (output power = 3 mW), a 10 mm pDNA-coated functionalized etched TFBG (ETFBG), a container and an OSA (resolution = 0.05 nm) for the detection of DNA, as depicted in Figure 2b. Each side of the modified ETFBG was fixed on the translational stage using an adhesive glue (LOCTITE 431, Henkel, Düsseldorf, Germany) to prevent a strain-induced shift in the transmission spectrum. The incident light beam excites the core and cladding modes in ETFBG and the interaction of cladding modes with the surrounding environment leads to the shift in transmitted ETFBG spectrum that can be monitored utilizing OSA.

The response of the bare ETFBG to varying RI of glycerine solution was investigated to ascertain its efficiency towards the ambient environment. Several solutions with RI ranging from 1.3328 to 1.3526 were prepared by mixing glycerine with DI water and the RI of each solution was determined using a handheld digital refractometer (Reichert AR200, New York, NY, USA). Figure 3a illustrates the spectral response of ETFBG to a change in the RI of the solutions at room temperature and the zoomed view of the red shift of one of the minima corresponding to cladding mode resonance is shown in Figure 3b. This type of movement can be attributed to the change in the effective index of cladding modes due to the interaction of the evanescent field of the cladding modes with the specific RI solution. On the other hand, the minima corresponding to Bragg resonance remain insensitive towards the change in RI of solutions, as displayed in Figure 3c. Therefore, this minimum can be utilized to detect temperature as well as maintain the straightness of the probe by monitoring the strain induced by the mechanical setup. It can be noted that the amplitude of the Bragg resonance dip is lower compared to conventional FBG because of the limited light propagation in the core. Repeating the above procedure under the same environmental conditions, another two probes were also fabricated, with their diameter being 41 μm (probe 1) and 38 μm (probe 3), which were slightly different that of from probe 2 (diameter = 40 μm). All these probes with different diameters were investigated for RI sensing (see Figure 3a,d,e). As can be observed from Figure 3f, the average sensitivity of probe 1, probe 2 and probe 3 was calculated to be 24.53 nm/RIU, 25.79 nm/RIU and 27.91 nm/RIU, respectively.

## 3. Materials and DNA Sensor Fabrication

### 3.1. Materials

Sulfuric acid (H_2_SO_4_), hydrogen peroxide (H_2_O_2_), phosphate-buffered saline (PBS 1X, pH 7.2–7.4), ethanol, polyethylenimine (PEI, 50 wt% aqueous solution, molecular weight = 20,000 g/mol), poly(acrylic acid) (PAA, 35 wt% aqueous solution, molecular weight = 100,000 g/mol), sodium hydroxide (NaOH) and hydrochloric acid (HCl) were purchased from Sigma Aldrich (St. Louis, MO, USA) to fabricate the DNA sensor. All of these reagents were of analytical grade and utilized without any additional purification. All the DNA sequences with 26 nucleotide bases were obtained from Sangon Biotech (Shanghai, China). The DNA sequences are pDNA (probe DNA): 5′-TCC AGA CAT GAT AAG ATA CAT TGA TG-3′; cDNA (complementary DNA): 5′-CAT CAA TGT ATC TTA TCA TGT CTG GA-3′; and non-cDNA (non-complementary DNA): 5′-CTC ACG TTA ATG CAT TTT GGT C-3′. The stock concentrations of DNA were prepared through a dilution process with PBS buffer to maintain the pH in the solution during detection. All the studies were conducted using deionized (DI) water with a resistivity of 18.2 MΩ cm [34].

### 3.2. Sensor Fabrication for DNA

The functionalization process of the sensor for DNA detection involves several steps, as depicted in Figure 4. Initially, the ETFBG was drenched in ethanol to remove the contaminants on the fiber surface. A piranha solution was prepared by blending H_2_O_2_ into H_2_SO_4_ at a ratio of 30%:70% (*v*/*v*) and the sensor was submerged in it for an hour followed by a wash with DI water to create a negatively charged hydroxyl bond (-OH) on its surface. Afterwards, the PEI/ PAA mixture was prepared by adding 10 mg/mL of PEI with 2 mg/mL of PAA at a ratio of 3:2 (*w*/*w*). The solution was ultrasonicated for 30 min to make it homogenous. Furthermore, the pH value of the PEI and PAA liquid was adjusted to pH 4 and pH 10 by 1 M HCL and 1 M NaOH, respectively. The scanning electron microscope (SEM) image shown in Figure 5 validates the PEI/PAA coating layer on the functionalized probe. The ETFBG modified with piranha solution was then thermally coated with a PEI/PAA composite at 75 °C for three hours, where the PEI and PAA provide positive and negative charges. The PEI is quickly affixed to the negatively charged ETFBG external due to the presence of amino nitrogen in its polymeric backbone, as it generates vastly positive charges [35]. On the contrary, the PAA functions as a sandwich chain structure that creates bonds with both negatively charged DNA and PEI through its positively charged carboxylic chain (COOH). Subsequently, the PEI/PAA-coated sensor was dipped into a 3 μM pDNA solution for an hour at room temperature to complete the fabrication process and immobilize the pDNA to the coated layer.

## 4. Results and Discussion

In this research, the spectrum corresponding to ETFBG coated with PEI/PAA (probe A, diameter = 40 μm) was recorded using the aforementioned experimental setup, as shown in Figure 6a. The coated fiber was then immersed in a PBS buffer solution for a few minutes to clean any additional contaminants from the fiber surface. Upon immersion, the spectrum exhibited a red shift due to the increase in the RI of the ambient environment as illustrated in Figure 6b. Subsequently, the PBS-treated probe was again dipped into a pDNA-based solution with 3 μM concentration and this led to the attachment of pDNA to the PEI/PAA coating layer. As can be seen in Figure 6b, the interference minima shifted towards the longer wavelength as the pDNA became immobilized. The probe was again washed with PBS to remove the unaffiliated pDNa, resulting in a blue shift of the transmission minima corresponding to cladding mode resonance. Afterwards, the pDNA-treated ETFBG was submerged into solutions of cDNA, with concentrations ranging from 1 μM to 3 μM to determine sensor sensitivity.

The sensor was kept in each solution for an hour to record the transmission spectrum. The resultant shift in the transmission spectrum is displayed in Figure 6b, while the zoomed view of the shift in transmission minima corresponding to different wavelengths is plotted in Figure 6c,d. The red shift of the minima in these figures can be attributed to the increase in the effective index of cladding modes due to the hybridization between pDNA and target cDNA. Simultaneously, the wavelength of the transmission minima corresponding to Bragg resonance remained unaffected during the change in concentration of the solutions as presented in Figure 6e. The zero shift of the Bragg resonance minima confirmed that the temperature of the ambient environment, as well as the strain, remained unchanged during the experiment, thereby demonstrating the reliability of the results.

The above investigations were repeated for probe B (diameter = 38 μm) and probe C (diameter = 36 μm) under identical conditions, and the results were plotted in Figure 6f–j and Figure 6k–o, respectively. The wavelength shift in transmission minima corresponding to all these probes due to changes in cDNA concentrations is summarized in Figure 7a–c. As can be seen in these figures, the sensitivity of probe A, probe B and probe C corresponding to two different transmission minima was found to be 260 pm/μM and 230 pm/μM, 320 pm/μM and 230 pm/μM and 286 pm/μM and 277 pm/μM, respectively. It can be easily noticed from the results that the transmission minima corresponding to higher-order cladding mode is more sensitive to DNA hybridization due to the higher spatial distribution of these modes in comparison to lower-order modes [36]. The slight variation in sensitivity could be attributed to the variation in diameter of the ETFBG and the slight variation in thickness of the PEI/PAA coating layer. Furthermore, the detection accuracy of the sensor with maximum sensitivity (probe B) was also investigated. The limit of detection (LOD) of the sensor was found to be 0.65 μM according to the following equation: LOD = 3σ/S, where σ (σ = 0.071 nm) and S refer to the standard deviation and sensitivity of probe B, respectively. The average quality factor (Q) and the figure of merit (FOM) were also calculated as 1.12 × 10^3^ and 0.24 utilizing the equation λ/∆λ_FWHM_ and SQ/λ, respectively, where λ is the resonance wavelength and ∆λ_FWHM_ is the full-width-half-maximum linewidth of the resonance wavelength. Furthermore, the performance of the proposed sensor compared with other DNA sensors is presented in Table 1.

The specificity of the prepared sensors was verified by submerging them in solutions of non-complementary DNA with a concentration of 1 μM for a period of one hour. The response of two of the minima corresponding to cladding modes were monitored and the results are shown in Figure 8a,b. As can be observed from this figure, the minima show a negligible shift, which verifies that no binding occurred between the coated probe and non-cDNA as opposed to cDNA. The wavelength shift corresponding to two minima of each probe was recorded every 5 min and the results are shown in Figure 8c–e where the shift is negligibly small. Therefore, the proposed probe can be utilized for cDNA detection with high accuracy and specificity.

## 5. Conclusions

A short-length TFBG-inscribed fiber optic sensor has been fabricated for DNA recognition using transmission-based interrogation. The sensor sensitivity towards the surrounding medium has been enhanced by etching the inscribed TFBG with HF for 30 min. The transmission minima corresponding to cladding mode resonance has been monitored to detect cDNA concentration, while the Bragg resonance has been utilized to monitor the error in measurement due to strain and temperature. The response of the etched sensors has been investigated to change in the RI of the surrounding medium to verify its potential for detection. The probes have been coated with a PEI/PAA layer using a thermal coating technique and pDNA was attached to this coating layer by dipping them in a 3 μM of solution of pDNA. Upon immersion of the PEI/PAA- and pDNA-treated probes in different concentrations of cDNA solutions, the wavelength corresponding to the minima showed a red shift and the maximum sensitivity was found to be 320 pm/μM, where the measurement range was from 0 μM to 3 μM. Furthermore, the sensors have been found to be unresponsive towards non-complementary DNA, thereby proving its selectivity. This type of sensor has the potential to identify DNA in several areas (e.g., medical, drug and genetic) due to its biocompatibility, robustness and low fabrication cost.

## Figures and Tables

**Figure 1 sensors-23-07019-f001:**
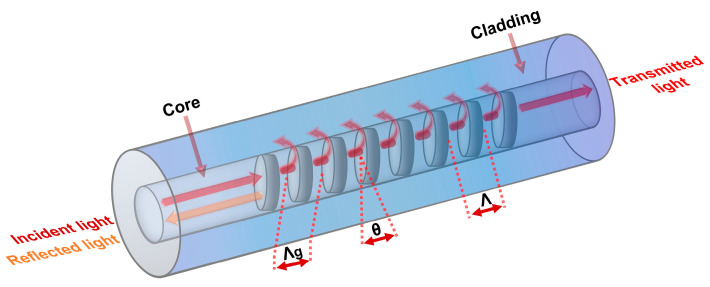
Schematic representation of a tilted fiber Bragg grating.

**Figure 2 sensors-23-07019-f002:**
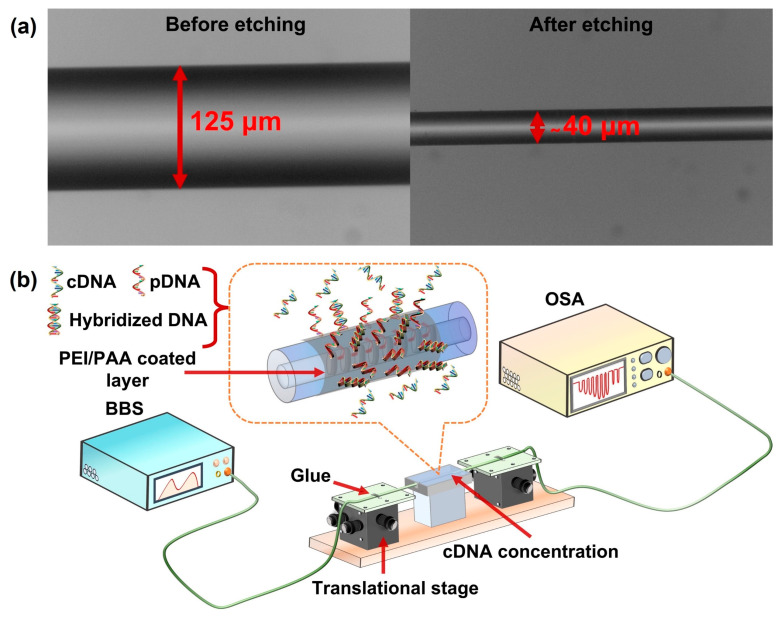
(**a**) Microscopic image of fiber before and after etching; (**b**) experimental setup for DNA detection.

**Figure 3 sensors-23-07019-f003:**
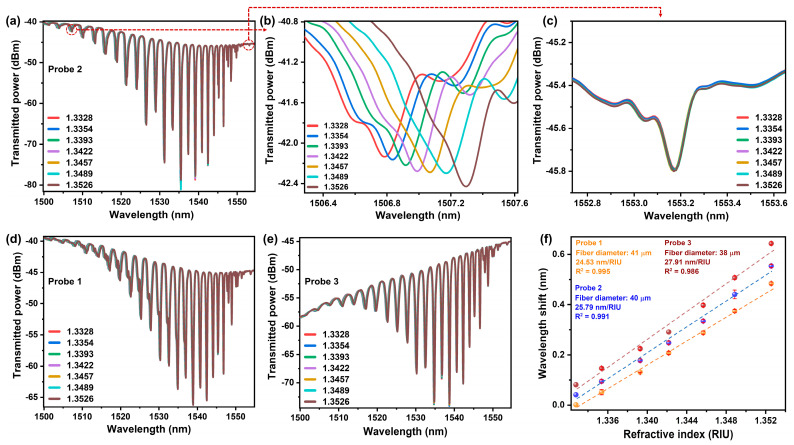
(**a**) shows the transmission spectrum of ETFBG corresponding to probe 2. (**b**,**c**) refer to the responses of minima corresponding to cladding mode resonance and Bragg resonance to changes in RI of solutions, respectively. (**d**,**e**) show the transmission spectrum of ETFBG corresponding to probe 1 and probe 3, respectively. (**f**) The average sensitivity of the three probes.

**Figure 4 sensors-23-07019-f004:**
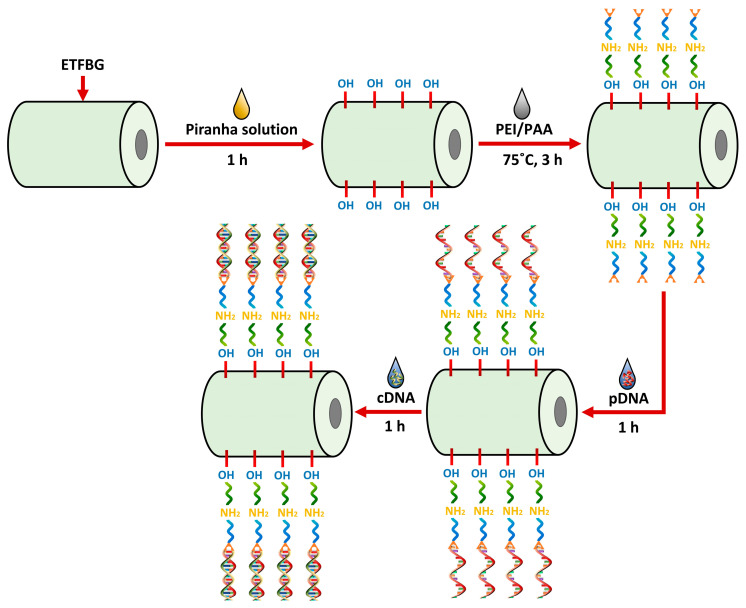
Schematic of the step by step procedure for surface modification of ETFBG for complementary DNA detection.

**Figure 5 sensors-23-07019-f005:**
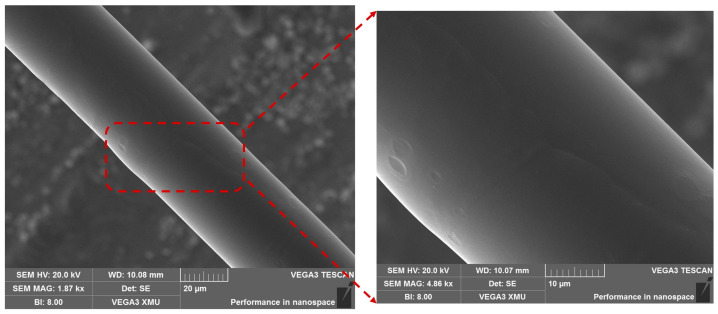
SEM illustration of the PEI/PAA-coated ETFBG probe.

**Figure 6 sensors-23-07019-f006:**
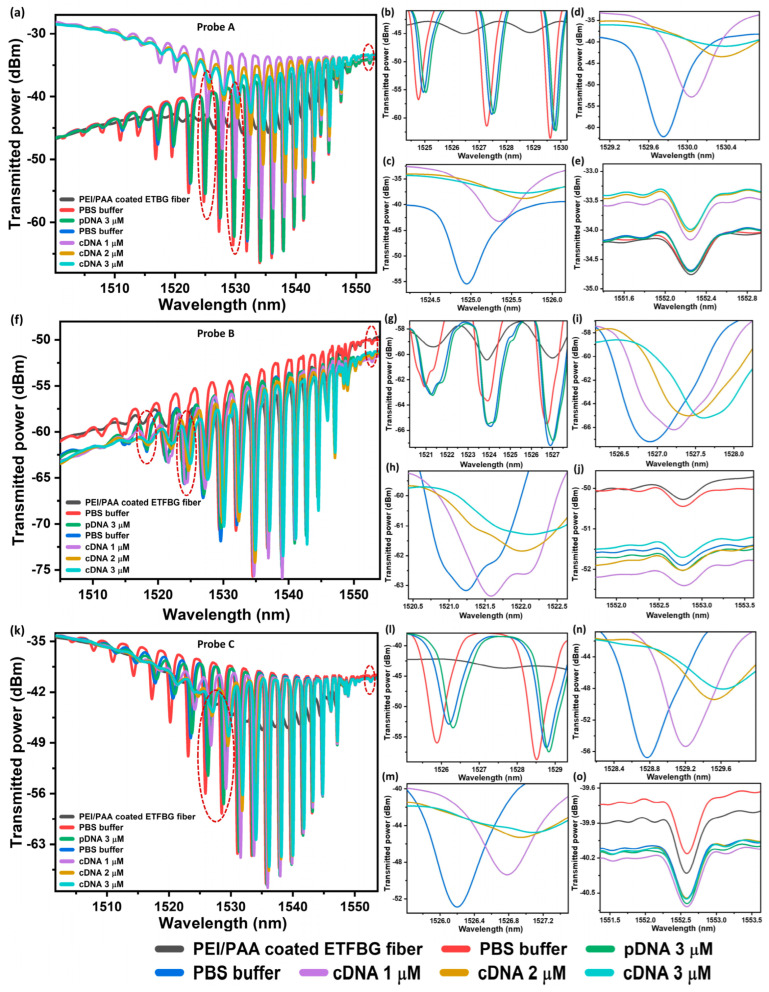
(**a**,**f**,**k**) show the response of transmission spectra of the three ETFBG probes to different concentrations of cDNA solutions. The inset (**b**–**e**), (**g**–**j**) and (**l**–**o**) refer to the core and cladding resonance in the transmission mode of probe 1, probe 2 and probe 3, respectively.

**Figure 7 sensors-23-07019-f007:**
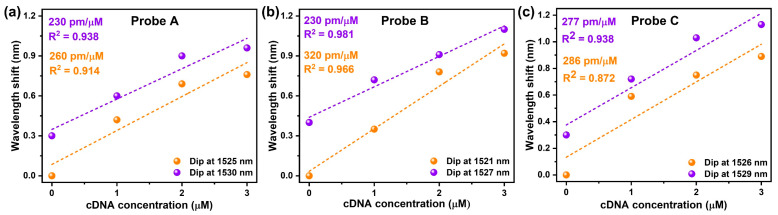
(**a**–**c**) Sensitivity of the three probes towards concentration of cDNA.

**Figure 8 sensors-23-07019-f008:**
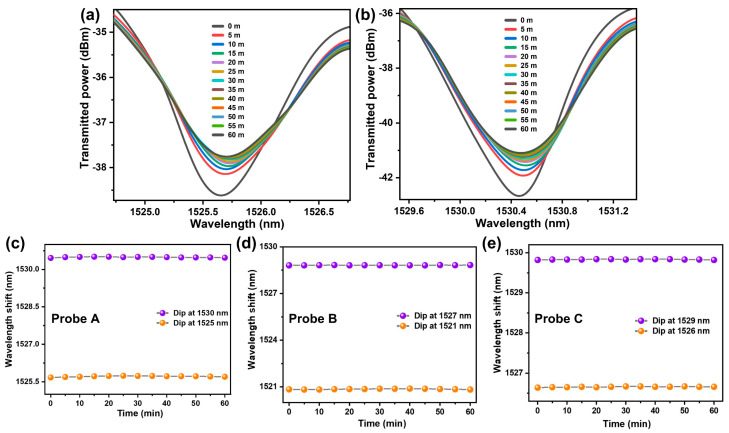
(**a**,**b**) Response of probe A towards non-cDNA. (**c**–**e**) represent the wavelength shift of transmission minima corresponding to the cladding mode resonance of three probes.

**Table 1 sensors-23-07019-t001:** Sensor performance in comparison with various DNA biosensors.

Sensing Mechanism	Optical Fiber Structure	Coating Materials	Detection Range	Sensitivity	LOD	Ref.
FPI	C-type fiber + SMF	Piranha solution/APTEs	1 μM	4310 pm/μM	67.5 nM	[19]
MZI	Tapered exposed core fiber + SMF	Piranha solution/APTEs	0–30 nM	61.8 pm/nM	0.31 nM	[20]
LPG	Side-polished fiber	PLL	1 μM	1182 pm/μM	-	[24]
LPG	SMF	APTEs/EDC	2 μM	245 pm/μM	4 nM	[25]
FBG	Microfiber	PLL	0.5–1 μM	456 pm/μM	0.5 μM	[26]
SPR + TFBG	SMF	Gr/Ag	0.0001–2 nM	-	3 pM	[31]
TFBG	Etched SMF	PEI/PAA	0–3 μM	320 pm/μM	0.65 μM	**This work**

## Data Availability

Not applicable.

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
