# Peer review of "Label-Free DNA Detection Using Etched Tilted Bragg Fiber Grating-Based Biosensor"

_sensors, 2023, doi:10.3390/s23167019_

Round 1
Reviewer 1 Report
The comments on the manuscript entitled "Label-free DNA detection using etched TFBG based biosensor" by
Abdullah Al Noman et al.:
The manuscript presents a label-free based fiber optic biosensor based on etched tilted Bragg fiber grating. The results reveal that the maximum sensitivity and detection limit are obtained 320 pm/μM and 0.65 μM, respectively. The topic is interesting and the results show good improvement, however the manuscript needs some revisions as follows:
1. Many grammatical and spelling mistakes are found. (For example, line 227: ...Figure 7a, 7b, and 7c, respectively. As can be these figures, the sensitivity of probe A,). The manuscript should be carefully edited.
2. Why did authors select solutions with RI ranging from 1.3328 to 1.3526? Please clarify.
3. What is the reason behind fabricating the probes with diameters of 41 µm, 40 µm, and 38 µm?
4. A comparison table for structure, sensing mechanism, performance parameters, and coating material between the proposed sensor and other DNA sensors is needed
5. It is recommended to complete the literature survey considering the following references: (a) Design of two-dimensional photonic crystal biosensor using DNA detection, Phosphorus, Sulfur, and Silicon and the Related Elements, 195, 2020; (b) Two-Dimensional photonic crystal Biosensors: A review, Optics and Laser Technology, 144, 2021; (c) Ultrasensitive DNA Detection Using Photonic Crystals, 10.1002/anie.200801998.
6. The results focus on the sensitivity and detection limit. Some important parameters of the biosensors such as quality factor should be defined and addressed.
7. It is suggested to define a figure of merit (FoM) for the proposed structure.
8. Is there any optimization in the physical parameters of the biosensor? Please explain.
Many grammatical and spelling mistakes are found. (For example, line 227: ...Figure 7a, 7b, and 7c, respectively. As can be these figures, the sensitivity of probe A,). The manuscript should be carefully edited.
Reviewer 2 Report
the article entitled (Label-free DNA detection using etched TFBG based biosensor) sounds good as a developed biosensor for sensing the DNA
but there are some comments:
1- the fiber etching process, the time, and concentration of HF, and the effect of concentration and time in the fiber size and index???
2-in SEM images, the bare fiber before PEI/PAA, and after immobilization of pDNA as well should be added
3- another characterization technique for the fiber with different layering should be included in the study (i.e. FTIR, UV-VIS)
4-3 μM of probe??? why this concentration??? did the authors study the coverage of the functionalized fiber surface????
5-the hybridization time (1 hour), did the authors study the incubation time?
6- Does the hybridization occur at room temperature or did the authors apply a certain incubation temperature? please clarify.
7- please clarify at which wavelength could you investigate your biosensor.
8- have the author compared the proposed biosensor with the other DNA optical biosensor to clarify their biosensor vintage
English language needs an extensive editing
Reviewer 3 Report
In this paper, the authors proposed a label-free fiber optic biosensor based on an etched tilted Bragg fiber grating (TFBG). The sensing probe achieves specific detection of cDNA. This article is clear, concise, and within the scope of the journal. However, some suggestions are provided:
1. The authors can show the É…g in figure 1.
2. What is the unit of 1300?
3. In Figure 3, the authors only provided the linear fitting results for probes 1 and 3. It is suggested that the authors also display the transmission spectra of probes 1 and 3.
4. On page 5, line 158, the letter 'l' in HCl should be lowercase. On pages 6, lines 174 and 175, the numbers in H2O2 and H2SO4 should be subscripted.
5. There should be a space between the number and the unit.
6. In Figure 6, it can be observed that the transmission spectra become broader at high concentrations of cDNA. Please explain the reason behind this. Additionally, there are differences in the transmission spectra of the three sensing probes at different concentrations, especially Probe B, which shows significant deviations compared to the other two probes. Please provide an explanation for this.
Reviewer 4 Report
Comments: This article reports Label-free DNA detection using etched TFBG-based biosensor. The structure of the synthesized materials has been characterized well and these analyses are reasonable. However, authors should address the following comments for its acceptance.
1. Numerous studies on similar materials have been reported in the literature, how is the synthesis method/work different or better than those reported earlier? Author should highlight this in the introduction part.
2. Author needs to supply the XRD pattern of before and after etched materials for a clear understanding of readers.
3. EDX spectra of PEI/PAA coated ETFBG probe to confirm chemical composition.
4. In order to show the superiority of the current materials, comparisons over the other related materials reported in the past literatures are necessary. Detection performances of the current materials have to be compared with those of the other materials and reasons for performance improvements have to be discussed.
5. The authors need to be double-checked the whole manuscript to eliminate syntax and format errors.
Minor spell check is needed.
Round 2
Reviewer 1 Report
The revised manuscript can be accepted for publication in Sensors, however, the reviewer suggests using "tilted Bragg fiber grating" instead of "TFBG" in the title.
Author Response
Yes. We agree with this suggestion and revised it accordingly.
Reviewer 2 Report
the authors figured all comments out
need an extensive editing
Author Response
Thank you for the reveiwer's comments again.
Reviewer 4 Report
The authors have moderately addressed the issues raised by the reviewers. Hence, the revised version of the manuscript may acceptable to the journal standard.
Spell check required.
Author Response

(The authors gave the same response as above.)
